# Hierarchical Normalization for Robust Monocular Depth Estimation

Chi Zhang[1], Wei Yin[2], Zhibin Wang[1], Gang Yu[1],[*], Bin Fu[1], Chunhua Shen[3]

[1]Tecent PCG, China    [2]DJI Technology, China    [3]Zhejiang University, China

[1] {johnczhang, brianfu, skicyyu}@tencent.com; [2] yvanwy@outlook.com; [3] Chunhua@icloud.com

## Abstract

In this paper, we address monocular depth estimation with deep neural networks. To enable training of deep monocular estimation models with various sources of datasets, state-of-the-art methods adopt image-level normalization strategies to generate affine-invariant depth representations. However, learning with the image-level normalization mainly emphasizes the relations of pixel representations with the global statistic in the images, such as the structure of the scene, while the fine-grained depth difference may be overlooked. In this paper, we propose a novel multi-scale depth normalization method that hierarchically normalizes the depth representations based on spatial information and depth distributions. Compared with previous normalization strategies applied only at the holistic image level, the proposed hierarchical normalization can effectively preserve the fine-grained details and improve accuracy. We present two strategies that define the hierarchical normalization contexts in the depth domain and the spatial domain, respectively. Our extensive experiments show that the proposed normalization strategy remarkably outperforms previous normalization methods, and we set new state-of-the-art on five zero-shot transfer benchmark datasets.

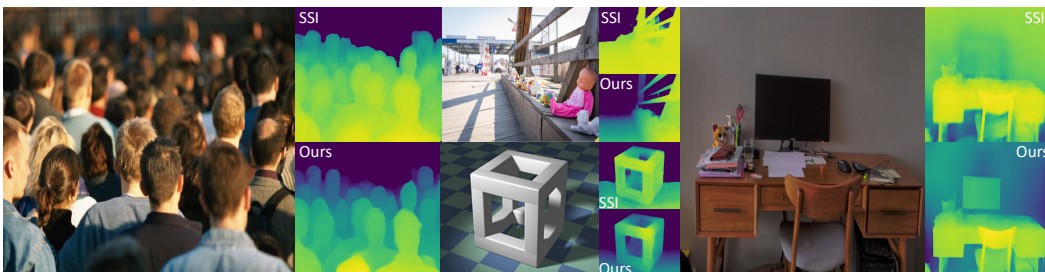

**Figure 1:** We propose a hierarchical depth normalization strategy to improve the training of monocular depth estimation models. Compared with the previous normalization strategy SSI [24], our design effectively improves the details and smoothness of predictions. We visualize the predictions of close regions with heat maps to observe details.

## 1 Introduction

Data-driven deep learning based monocular depth estimation has gained wide interest in recent years, due to its low requirements of sensing devices and impressive progress. Among various learning

---

[*]Corresponding author

36th Conference on Neural Information Processing Systems (NeurIPS 2022).

objectives of deep monocular estimation, zero-shot transfer carries the promise of learning a generic depth predictor that can generalize well across a variety of scenes. Rather than training and evaluation on the subsets of individual benchmarks that usually share similar characteristics and biases, zero-shot transfer expects the models to be deployed for predictions of any in-the-wild images.

To achieve this goal, large-scale datasets with equally high diversity for training are necessary to enable good generalization. However, collecting data with high-quality depth annotations is expensive, and existing benchmark datasets often show limitations in scales or diversities. Many recent works [24, 37] seek mix-dataset training, where datasets captured by various sensing modalities and in diverse environments can be jointly utilized for model training, which largely alleviates the difficulty of obtaining diverse annotated depth data at scale. Nevertheless, the mix-data training also comes with its challenges, as different datasets may demonstrate inconsistency in depth representations, which causes incompatibility between datasets. For example, the disparity map generated from web stereo images [32] or 3D movies [24] can only provide depth annotations up to a scale and shift, due to varied and unknown camera models.

To solve this problem, state-of-the-art methods [24, 37] seek training objectives invariant to the scale-and-shift changes in the depth representations by normalizing the predictions or depth annotations based on statistics of the image instance, which largely facilitates the mix-data learning of depth predictors. However, as the depth is represented by the magnitude of values, normalization based on the instance inevitably squeezes the fine-grained depth difference, particularly in close regions. Suppose that an object-centric dataset with depth annotations is available for training, where the images sometimes include backgrounds with high depth values. Normalizing the depth representations with global statistics can distinctly separate the foreground and background, but it may meanwhile lead to an overlook of the depth difference in objects, which may be our real interest. As the result, the learned depth predictor often excels at predicting the relative depth representations of each pixel location with respect to the entire scene in the image, such as the overall scene structure, but struggles to capture the fine-grained details.

Motivated by the bias issue in existing depth normalization approaches, we aim to design a training objective that should have the flexibility to optimize both the overall scene structure and fine-grained depth difference. Since depth estimation is a dense prediction task, we take inspiration from classic local normalization approaches, such as local response normalization in AlexNet [16], local histogram normalization in SIFT [20] and HOG features [6], and local deep features in DeepEMD [40], which rely on normalized local statistics to enhance local contrast or generate discriminative local descriptors. By varying the size of the local window, we can control how much context is involved to generate a normalized representation. With such insights, we present a hierarchical depth normalization (HDN) strategy that normalizes depth representations with different scopes of contexts for learning. Intuitively, a large context for normalization emphasizes the depth difference globally, while a small context focuses on the subtle difference locally. We present two implementations that define the multi-scale contexts in the spatial domain and the depth domain, respectively. For the strategy in the spatial domain, we divide the image into several sub-regions and the context is defined as the pixels in individual cells. In this way, the fine-grained difference between spatially close locations is emphasized. By varying the grid size, we obtain multiple depth representations of each pixel that rely on different contexts. For the strategy in the depth domain, we group pixels based on the ground truth depth values to construct contexts, such that pixels with similar depth values can be differentiated. Similarly, we can change the number of groups to control the context size. By combining the normalized depth representations under various contexts for optimization, the learner emphasizes both the fine-grained details and the global scene structure, as shown in Fig. 1.

To validate the effectiveness of our design, we conduct extensive experiments on various benchmark datasets. Our empirical results show that the proposed hierarchical depth normalization remarkably outperforms the existing instance-level normalization qualitatively and quantitatively. Our contributions are summarized as follows:

- We propose a hierarchical depth normalization strategy to improve the learning of deep monocular estimation models.
- We present two implementations to define the multi-scale normalization contexts of HDN based on spatial information and depth distributions, respectively.
- Experiments on five popular benchmark datasets show that our method significantly outperforms the baselines and sets new state-of-the-art results.

Next we review some works that are closest to ours and then present our method in Section 3. In Section 4, we empirically validate the effectiveness of our method on several public benchmark datasets.

## 2 Related Work

**Deep monocular depth estimation.** As opposed to early works on monocular depth estimation based on hand-crafted features, recent studies advocate end-to-end learning based on deep neural networks. Since the pioneer work [7] first adopts deep neural networks to undertake monocular depth estimation, significant progress has been made from many aspects, such as network architectures [17, 22, 18], large-scale and diverse training datasets [34, 38], loss functions [24, 37], multi-task learning [39], synthesized dataset [8], geometry constraint [34, 22, 35, 36] and various sources of supervisions [33, 37].

For supervised training, collecting high-diversity data with ground-truth depth annotations at scale is expensive. Recent works based on ranking loss [2, 33] and scale-and-shift invariant losses [24, 37] enable network training with other forms of annotations, such as ordinal depth annotations [2, 32] , or relative inverse depth map [33, 34, 24] generated by uncalibrated stereo images using optical flow algorithms [28]. In particular, scale-and-shift invariant (SSI) loss [24] and image-level normalization loss [37] allow data from multiple sources to be learned in a fully supervised depth regression manner, which largely facilitates large-scale training and improves the generation ability of learning based depth estimators. The SSI loss removes the major incompatibility between various datasets, *i.e.*, the scale and shift changes, by transforming the depth representation into a canonical space through normalization. With such advances, zero-shot transfer is made possible, where the network learned on a large-scale database with high diversity can be directly evaluated on various benchmarks without seeing their training samples, which is the focus of this paper.

Furthermore, some literature [1, 44, 10, 26] proposes to solve the monocular depth estimation problem without sensor-captured ground truth but leverages the training signal from consecutive temporal frames or stereo videos. However, most of these methods need the camera intrinsic parameters for supervision.

**Normalization in CNNs.** Normalization is widely adopted in deep neural networks, while different normalization strategies are employed for different purposes. For instance, batch normalization (BN) [13] normalizes the feature representations along the batch dimension to stabilize training and accelerate convergence. BN usually prefers large normalization contexts to obtain robust feature representation. On the other hand, another line of normalization methods relies on local statistics. For example, instance normalization [29] and its variants [12, 21] based on instance-level statistics dominate the style transfer task, as they emphasize the unique styles of individual images. A collection of literature seeks normalization in local regions. For example, the well-known SIFT feature [20] and HOG features [6] are based on the normalized local statistics to generate discriminative local features. DeepEMD [40, 41] computes the optimal transport between local normalized deep features as a distance metric between images. Since the local details and the overall scene structure are both important for a depth estimator, we incorporate the ideas of both global normalization and local normalization in the monocular depth estimation models.

## 3 Method

In this section, we first briefly summarize the preliminaries of the task. Then we define a unified form of depth normalization and show that the normalization strategy in scale-and-shift invariant loss[24] is a special case. Finally, we present two implementations of our proposed hierarchical depth normalization approaches based on the spatial domain and the depth domain, respectively.

### 3.1 Preliminaries

We aim to boost the performance of zero-shot monocular depth estimation with diverse training data. In our pipeline, we input a single RGB image $I \in \mathbb{R}^{H \times W \times 3}$ to the depth prediction network $\mathcal{F}(\cdot)$ to generate a depth map $D \in \mathbb{R}^{H \times W \times 3}$. Instead of directly regressing the output map with the ground-truth depth supervision, state-of-the-art methods [34, 24, 37] normalize the depth representations

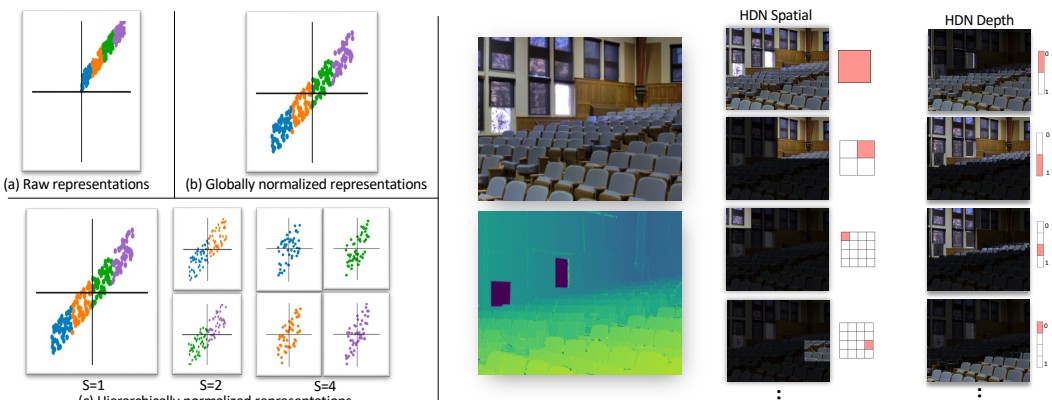

**Figure 2: Left**: An illustration of different normalization strategies. Given the raw representations (a), previous methods (b) normalize the representations globally. Our proposed hierarchical depth normalization (c) strategy divides representations into $S$ groups, denoted by different colors, and apply normalization to individual groups to generate representations. We select different $S$ to obtain multiple representations for learning. By doing so, we can emphasize both the global data distribution and the fine-grained difference between representations during training. **Right**: Our two normalization strategies for the depth estimation task that group pixels in the spatial domain (HDN-Spatial) and the depth domain (HDN-Depth), respectively. The HDN-Spatial groups pixels in sub regions, while the HDN-Depth groups pixels based on depth distributions. The red cells denote the selected pixels for normalization.

before computing the regression loss. In our work, we focus on the depth normalization part, which means our methods can be easily combined with state-of-the-art network structures or loss functions.

## 3.2 Unified Form of Depth Normalization

Let $\mathbf{d} \in \mathbb{R}^M$ and $\mathbf{d}^* \in \mathbb{R}^M$ denote the vectorized predicted depth map and the ground truth depth map, respectively, where $M$ is the number of pixels with valid annotations. Our goal is to generate the normalized representations, $\mathcal{N}_{u_i}(d_i)$ and $\mathcal{N}_{u_i}(d_i^*)$ for computing the regression loss of each location $i$, where $u_i$ is the set of location indexes that constitute the context of location $i$, and $\mathcal{N}_{u_i}$ is the normalization function based on the context $u_i$. For example, the normalization employed in the scale-and-shift invariant loss (SSI) [24] is to explicitly remove the estimated scale and shift from the raw depth representations:

$$\mathcal{N}_{u_i}(d_i) = \frac{d_i - \mathtt{median}_{u_i}(\mathbf{d})}{\frac{1}{|u_i|} \sum_{j=1}^{|u_i|} |d_i - \mathtt{median}_{u_i}(\mathbf{d})|}, \quad \mathcal{N}_{u_i}(d_i^*) = \frac{d_i^* - \mathtt{median}_{u_i}(\mathbf{d}^*)}{\frac{1}{|u_i|} \sum_{j=1}^{|u_i|} |d_i^* - \mathtt{median}_{u_i}(\mathbf{d}^*)|}, \quad (1)$$

where the $\mathtt{median}_{u_i}$ operator computes the median depth of locations that belong to $u_i$. The instance-level normalization in SSI assigns a global context $u_{\mathrm{glo}}$ for each pixel that involves all pixel locations with valid depth annotations, *i.e.*, $u_j = u_{\mathrm{glo}}, \forall j \in u_{\mathrm{glo}}$. Finally, the SSI loss $\mathcal{L}_{u_i}^{\mathrm{SSI}}$ computes the mean absolute error between the normalized prediction and the ground truth, and losses over all locations are averaged to obtain the final loss $\mathcal{L}^{\mathrm{SSI}}$:

$$\mathcal{L}^{\mathrm{SSI}} = \frac{1}{M} \sum_{i=1}^{M} \mathcal{L}_i^{\mathrm{SSI}}, \text{where } \mathcal{L}_i^{\mathrm{SSI}} = |\mathcal{N}_{u_i}(d_i) - \mathcal{N}_{u_i}(d_i^*)| \quad (2)$$

Intuitively, a large normalization context that covers a large scope of locations emphasizes the relative depth representations with the global statistics, while a small context emphasizes the depth difference more locally. Based on this observation, we propose to assign multiple contexts at different scales for each location.

## 3.3 Hierarchical Depth Normalization

We next present two strategies that define the multi-scale contexts in the spatial domain and the depth domain, respectively, which are illustrated in Fig. 2.

**In the spatial domain (HDN-S).** Since images have 2D grid-like representations, we can define the context based on spatial positions. In this way, the context corresponds to a group of pixels in the sub-region. Defining context in spatial domain based on grids was also seen in segmentation models, where poolings are used to explicitly enlarge the effective receptive fields of features[43]. Here we adopt a similar strategy that evenly divides the image plane into several sub-regions based on a $S \times S$ grid. Then the pixels belonging to the same region share the same context for normalization. By varying the grid size, we can obtain multiple contexts for each pixel location. In our design, we select the grid size $S_{\text{spatial}}$ from $\{2^0, 2^1, 2^2, ...\}$ and apply normalization at each level. For example, at the highest level, the normalization is essentially the instance-level normalization that covers all pixels as the context, while at the lowest level, the normalization is limited to an $\frac{H}{S} \times \frac{W}{S}$ image patch.

**In the depth domain (HDN-D).** The other way to group close pixels is based on their ground-truth depth distributions, such that pixels with similar depth values share the same context. Here we present two strategies. The first one, denoted by *HDN-DP*, is to sort all the pixels based on their ground-truth depth values and evenly divide them into $S$ bins, where the numbers of pixels in each bin are $\frac{M}{S}$. The second strategy, denoted by *HDN-DR*, is to evenly divide the depth range presented in an image into $S$ bins, and classify pixels into different bins. Pixels belonging to the same bin share the same context. Since pixels with similar depth values may not be spatially close, long-range spatial dependency can be established by the shared contexts to promote global coherency. The bin number $S_{\text{depth}}$ is also selected from $\{2^0, 2^1, 2^2, ...\}$.

Suppose that the multi-scale contexts of location $i$ constitutes a set $U_i$, we compute the SSI losses generated by each context from $U_i$ and average them to obtain the final loss $\mathcal{L}^{\text{HDN}}$:

$$\mathcal{L}^{\text{HDN}} = \frac{1}{M} \sum_{i=1}^{M} \mathcal{L}_i^{\text{HDN}}, \text{where } \mathcal{L}_i^{\text{HDN}} = \frac{1}{|U_i|} \sum_{u \in U_i} |\mathcal{N}_u(d_i) - \mathcal{N}_u(d_i^*)|. \tag{3}$$

As we can see, by combing the contexts at different scales, the proposed learning objective emphasizes the depth difference at different levels, which can preserve the local fine-grained details while maintaining structural consistency. In contrast, the SSI loss is a special case that only focuses on the global structure.

We can also define the hierarchical normalization contexts based on irregular patterns, *e.g.*, segmentation masks. In fact, our design provides a useful way to explicitly utilize semantic knowledge for improving predictions. For example, if we define a normalization context that covers stuff and objects based on panoptic segmentation, it emphasizes the spatial relations between different instances and stuff. On the other hand, if the normalization context is defined based on the instance masks, we can emphasize the fine-grained depth difference in object appearance, such as the structures of vehicles. As we aim to design a generic depth normalization algorithm, we choose to define the hierarchical normalization based on regular patterns without seeking extra knowledge in this paper.

**Implementation.** Our proposed hierarchical normalization strategies are lightweight and easy to implement. Since the SSI loss functions are usually implemented as a function of ground truth depth maps, predicted depth maps, and valid-pixel maps, our approaches can be easily implemented by only modifying the valid-pixel maps to obtain batched computations, where the pixels out of the context are masked out.

## 4 Experiments

**Datasets** We follow LeReS [37] to construct a mix-data training set, which includes 114K images from Taskonomy dataset, 121K images from DIML dataset, 48K images from Holopix50K, and 20K images from HRWSI [33]. We withhold 1K images from all datasets for validation during training. The details of each dataset are as follows.

*Taskonomy.* Taskonomy [38] is a high-quality and large-scale dataset, which contains over 4 million images of indoor scenes from about 600 buildings. It includes annotations for over 20 tasks. In our experiments, we sampled around 114k RGB-D pairs for training.

*DIML.* DIML [14] contains synchronized RGB-D frames from Kinect v2 or Zed stereo camera. For the outdoor split, they are mainly captured by the calibrated stereo cameras. It contains various outdoor places, *e.g.*, offices, rooms, dormitory, exhibition center, street, road and so on. We used GANet [42] to recompute the disparity and depth maps for training.

| Norm. Method | DIODE | ETH3D | KITTI | NYU | ScanNet | Mean |
|---|---|---|---|---|---|---|
| | | | AbsRel↓ | | | Improv. |
| Instance | 47.1 | 19.4 | 13.2 | 9.6 | 10.22 | - |
| Batch | 45.0 (−4%) | 25.2 (+30%) | 14.9 (+12%) | 11.5 (+19%) | 12.2 (+19%) | (+15%) |
| Local-S | 33.1 (−30%) | 14.5 (−25%) | 19.3 (+45%) | 11.8 (+22%) | 11.5 (+12%) | (+5%) |
| Local-DR | 25.7 (−45%) | 12.7 (−35%) | 14.7 (+11%) | 9.4 (−2%) | 9.9 (−3%) | (15%) |
| Local-DP | 28.8 (−39%) | 14.4 (−26%) | 23.3 (+76%) | 10.4 (+8%) | 10.95 (+7%) | (+5%) |
| **HDN-S** | 36.4 (−22%) | 14.5 (−25%) | 12.5 (−6%) | 8.8 (−8%) | 9.5 (−6%) | (−14%) |
| **HDN-DR** | 24.9 (−47%) | 12.7 (−35%) | 11.9 (−10%) | 8.7 (−10%) | 9.4 (−8%) | (−22%) |
| **HDN-DP** | 25.1 (−47%) | 12.8 (−34%) | 13.2 (0%) | 8.7 (−9%) | 9.3 (−8%) | (−19%) |

**Table 1:** Comparison of different depth normalization strategies. Our proposed HDN outperforms the instance-level normalization baseline remarkably. Relying only on local normalization produces unstable results across benchmarks.

*HRWSI and Holopix50k.* HRWSI [33] and Holopix50k [11] are both diverse relative depth datasets. Although they contain diverse scenes and various camera settings, their provided stereo images are uncalibrated. Therefore, the stereo matching methods are inapplicable to recover the metric depth information. We use RAFT [28] to recover their relative depth representations for training.

**Implementation details.** We use the state-of-the-art monocular depth estimation network DPT-Hybrid [23] in our experiments. The input image size is $480 \times 480$ at both training and testing time. For evaluation on benchmarks, we re-size the outputs to the raw image resolutions with the bilinear interpolation. Random horizontal flip and random crop are employed for data augmentation. The default random crop size in all experiments is sampled in $[0.5, 1]$ of the raw image size, with the aspect ratio restricted in $[3/4, 4/3]$, and the random cropped patches are re-sized to the input resolution for training. We use the Adam optimizer with a learning rate of $10^{-5}$. The model is trained on 8 V100 GPU with the batch size of 32. In each mini-batch, we sample the equal number of images from different training data sources. For HDN in the spatial domain, we select the grid size $S_{\text{spatial}}$ from $\{2^0, 2^1, 2^2, 2^3\}$ to construct the hierarchical contexts. For HDN in depth domain, the group number $S_{\text{depth}}$ is chosen from $\{2^0, 2^1, 2^2\}$.

**Evaluation Metrics**. We include 5 popular benchmarks that are unseen during training, including, DIODE [30], ETH3D [25], KITTI [9], NYU [27], and ScanNet [5]. We follow the previous works to evaluate our model on zero-shot cross-dataset transfer. We use the mean absolute value of the relative error (AbsRel): $(1/M) \sum_{i=1}^{M} |d_i - d_i^*|/d_i^*$ and the percentage of pixels with $\delta_1 = \max(\frac{d_i}{d_i^*}, \frac{d_i^*}{d_i}) < 1.25$. Following Midas [24] and LeReS [37], we align the predictions and ground truth in scale and shift before evaluation. Please refer to our supplementary material for more experiment results and analysis.

### 4.1 Analysis

For ablation study and analysis, we sample a subset of 16K images evenly from our four training sets, and the input image size is $384 \times 384$.

**Comparison of normalization strategies.** At the beginning, we compare different normalization strategies. The first baseline is the instance-level normalization, *i.e.*, scale-and-shift invariant loss. We also include the **batch**-level depth normalization for a comprehensive study. For our proposed HDN, we report the results of models based on the spatial domain (**HDN-S**) and the depth domain (**HDN-DP** and **HDN-DR**). We also take the finest normalization level of the proposed HDN to validate whether local contexts alone help the training, which is denoted by **Local**. A detailed record of the experiment results is presented in Table 1.

As we can see, normalization along the batch dimension yields the worst results, which means that the scale-and-shift changes between data can not be addressed by utilizing batch-level statistics. Our proposed HDNs in both the depth domain and the spatial domain remarkably outperform the instance-level normalization baseline, which validates the effectiveness of our design. The HDNs in the depth domain are more effective. In particular, HDN-DR reduces the error by an average of 22% on the five benchmarks, so we use HDN-DR as the default model in rest experiments. We find that the instance normalization baseline performs poorly on DIODE and ETH3D datasets, particularly in

| Loss | AbsRel↓ | $\delta_1$ ↑ |
|------|---------|--------------|
| L1 | 14.7 | 80.1 |
| L1 + SSI [24] | 14.6 (−0.6%) | 79.1 |
| L1 + HDN | 13.3 (−9.5%) | 82.8 |

**Table 2:** Experiments results under the fully supervised setting. The experiment is conducted on NYU V2 dataset [27]. Our design can also improve the training of monocular estimation models under the fully supervised setting.

their outdoor splits. This may be caused by dataset bias or the shortage of diversity in the training datasets. In contrast, the proposed HDNs perform well on all benchmark datasets and even *reach state-of-the-art with only 16K training data, which are dozens of times fewer than the training sets used in recent methods.* Relying solely on local normalization, *i.e.*, the finest normalization level, can improve the baseline on some benchmarks, *e.g.*, DIODE and ETH3D, but it may meanwhile cause significant performance drops on other benchmarks, *e.g.*, KITTI. This suggests that the combination of local normalization and global normalization is the optimal strategy for robust training and good generalization.

**Delving into HDN.** We next study the characteristics of the proposed HDN. As the key difference between HDNs and instance-level normalization is the extra local normalization contexts, we first validate whether our methods still work well when different strength of random cropping is applied for data augmentation. We select the lower bound of cropping size from $\{1/8, 1/4, 1/2\}$ of the raw size to observe the improvements. We then gradually add more fine-grained normalization context levels and observe the performance changes. For example, if three levels are adopted for HDN-Depth, the group number $S_{\text{depth}}$ is chosen from $\{2^0, 2^1, 2^2\}$. When only the top level is adopted, the HDN degrades to the instance-level normalization baseline. We also report performance changes of our method relying on the finest levels alone, which is also the **Local** model variants in Table 1. For all experiments, we report the mean relative error drop rate (AbsRel) over the instance-level normalization baseline in Table 1, whose lower bound of random crop is 0.5.

The results are presented in Fig. 3. Removing the random crop (**No Crop**) harms the overall performance, which can be seen from the starting point of different charts that denote the instance-level normalization baselines. Although instance-level normalization with aggressive random crop (**Crop 1/8** and **Crop 1/4**) can also enforce local spatial normalization contexts, it may meanwhile lose useful contexts for feature encoding and causes performance drops. Our proposed HDNs can consistently and significantly outperform the baselines under various augmentation strengths, which indicates that data augmentation with random cropping and our HDNs are complementary.

For our HDNs, adding more fine-grained normalization context levels can not continually improve the performance. By observing the results of normalization relying on the local contexts alone, we find that they can still outperform the instance-level normalization baseline when the appropriate level is assigned. However, the performance is likely to get worse significantly when the contexts become too local, as can be seen from the result of **Local-DP**.

**The effectiveness of HDN under the fully supervised setting.** We next validate whether our proposed HDN can help the model training under the fully supervised setting, where the metric depth annotations are provided in the training set. We validate our design on the popular NYU V2 dataset, which contains 795 training images in Eigen split [7], by adding our proposed HDN loss as an auxiliary loss to a standard L1 regression loss. The result is shown in Table 2. Our proposed HDN can still effectively improve the performance, while the SSI baseline can hardly boost the performance, which shows the generalization capability of our methods in monocular depth estimation tasks.

**Qualitative results.** We present some qualitative comparisons in Figure 4. We mainly compare with the instance-level normalization baseline, *i.e.*, the scale-shift-invariant (SSI) loss [24]. All comparison cases are sampled from the zero-shot testing datasets. We can observe that our hierarchical normalization methods make better predictions in the smoothness of object surface and the sharpness of edges. Therefore, compared to the global instance-level normalization, our extra local normalization in the depth domain or the depth domain can force the network to master better knowledge of local geometries. Furthermore, we randomly sample 2K pixels from each image and plot the predicted depth value and ground truth depth value in the last two columns. Ideally, the

| Norm. Method | $\varepsilon_{\text{DBE}}^{\text{acc}}\downarrow$ | $\varepsilon_{\text{DBE}}^{\text{comp}}\downarrow$ | AbsRel$\downarrow$ |
|---|---|---|---|
| Instance-level | 2.55 | 5.84 | 9.81 |
| **HDN-S** | 2.35 | 5.15 | 8.58 |
| **HDN-DR** | 2.41 | 5.57 | 8.76 |
| **HDN-DP** | 2.47 | 5.67 | 8.99 |

**Table 3:** Results on iBims-1 dataset [15]. The proposed normalization methods outperform the baseline in metrics for evaluating the depth boundaries.

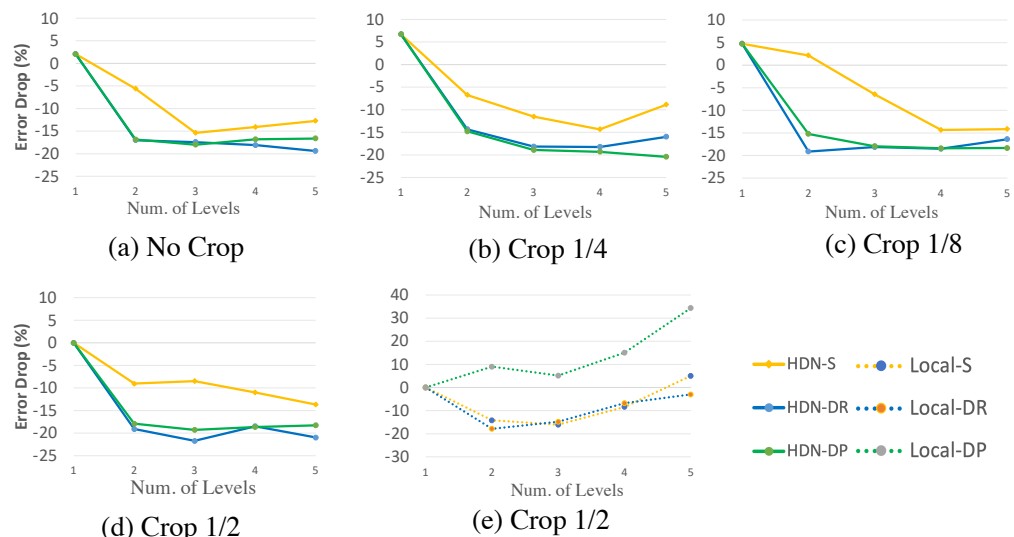

(a) No Crop       (b) Crop 1/4       (c) Crop 1/8

(d) Crop 1/2       (e) Crop 1/2

**Figure 3:** Different numbers of context levels for the proposed Hierarchical Depth Normalization and the variants based solely on the lowest level (Local). Lower means better. The proposed normalization approaches consistently outperform the baselines (number of level = 0) under various data augmentation strengths. Relying only on local normalization with very small contexts may harm the performance.

predicted affine-invariant depth should be strictly linear to the ground truth, *i.e.*, the red diagonal line. We can observe that the linearity of our method is much better than the baseline. Note that even if the baseline could achieve similar prediction accuracy sometimes (such as the second example), our methods generate more find-grained details.

**Evaluation of depth boundaries.** We have demonstrated the advantages of our proposed methods in generating sharper edges and better details through visualization. We next quantitatively evaluate the quality of edges based on experiments on iBims-1 dataset [15]. In particular, we choose the metrics specifically used for evaluating depth boundaries, *i.e.*, the edges. Please refer to [15] for detailed definitions of the metrics. The result is presented in Table 3. As we can see, our proposed normalization strategies outperform the baselines.

## 4.2 Comparison with State-of-the-Art

Finally, we compare our depth estimator (HDN-DR) with the state-of-the-art methods on five benchmark datasets, including, DIODE [30], ETH3D [25], KITTI [9], NYU [27], and ScanNet [5], which are unseen during training. As the result shows in Table 4, our method outperforms previous methods by a large margin on multiple benchmarks with fewer training data than recent state-of-the-art methods, such as LeReS [37] and Midas [24].

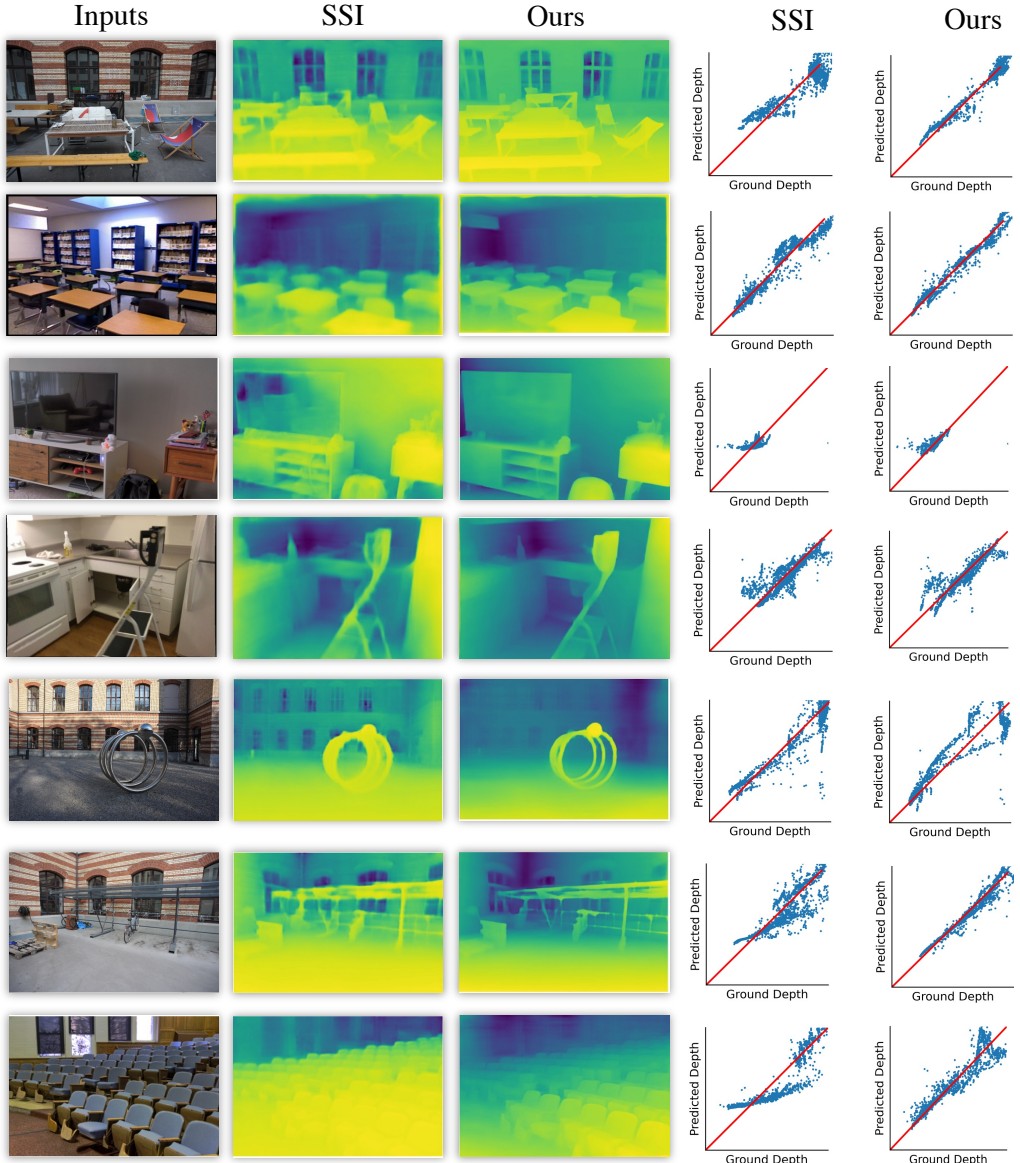

**Figure 4:** Qualitative comparison of our method and the baseline replying on instance-level normalization (SSI [24]). The red diagonal lines denote the ground-truth, and closer to it means better. Our method generates smoother surfaces, sharper edges and more details.

## 5 Limitations

We find that for datasets with very high image resolutions, such as ETH3D with $4032 \times 6048$ raw resolutions, a relatively small inference resolution, *e.g.*, $384 \times 384$, is worse than a relatively large resolution, *e.g.*, $800 \times 800$, but further increasing the resolution can not consistently improve the performance. Therefore, there is a lack of principle for setting the optimal input resolutions at test time. We believe that research on adaptive input resolutions and learnable test-time augmentations would be promising solutions in the future.

Our future work also includes better strategies to define the hierarchical normalization contexts, such as utilizing cross-domain knowledge and extra knowledge, *e.g.*, segmentation maps.

| Method | Training Data | NYU | | KITTI | | DIODE | | ScanNet | | ETH3D | |
|---|---|---|---|---|---|---|---|---|---|---|---|
| | | AbsRel↓ | $\delta_1$↑ | AbsRel↓ | $\delta_1$↑ | AbsRel↓ | $\delta_1$↑ | AbsRel↓ | $\delta_1$↑ | AbsRel↓ | $\delta_1$↑ |
| OASIS [4] | ∼ 140K | 21.9 | 66.8 | 31.7 | 43.7 | 48.4 | 53.4 | 19.8 | 69.7 | 29.2 | 59.5 |
| MegaDepth [19] | ∼ 150K | 19.4 | 71.4 | 20.1 | 66.3 | 39.1 | 71.2 | 26.0 | 64.3 | 39.8 | 52.7 |
| Xian et al.[33] | ∼ 30K | 16.6 | 77.2 | 27.0 | 52.9 | 42.5 | 61.8 | 17.4 | 75.9 | 27.3 | 63.0 |
| WSVD [31] | ∼ 1.5M | 22.6 | 65.0 | 24.4 | 60.2 | 35.8 | 63.8 | 18.9 | 71.4 | 26.1 | 61.9 |
| Chen et al.[3] | ∼ 795K | 16.6 | 77.3 | 32.7 | 51.2 | 37.9 | 66.0 | 16.5 | 76.7 | 23.7 | 67.2 |
| DiverseDepth [34] | ∼ 300K | 11.7 | 87.5 | 19.0 | 70.4 | 37.6 | 63.1 | 10.8 | 88.2 | 22.8 | 69.4 |
| MiDaS [24] | ∼ 2M | 11.1 | 88.5 | 23.6 | 63.0 | 33.2 | 71.5 | 11.1 | 88.6 | 18.4 | 75.2 |
| Leres [37] | ∼ 360K | 9.0 | 91.6 | 14.9 | 78.4 | 27.1 | 76.6 | 9.5 | 91.2 | 17.1 | 77.7 |
| Ours | ∼ 300K | **6.9** | **94.8** | **11.5** | **86.7** | **24.6** | **78.0** | **8.0** | **93.9** | **12.1** | **83.3** |

**Table 4:** Comparison with state-of-the-art methods on six zero-shot transfer benchmark datasets. Our model significantly outperforms previous methods and sets new state-of-the-art in many benchmarks. We use the MiDaS model reported in their original paper for comparison.

## 6   Conclusion

In this paper, we have presented a novel hierarchical depth normalization strategy for monocular depth estimation tasks. Compared with the existing instance-level normalization strategy that mainly focuses on the global structure of the scene in the image, our normalization and loss preserve both the fine-grained details and overall structure. Extensive experiments validate the effectiveness of our two implementations in the spatial domain and the depth domain, and new state-of-the-art performance is set on multiple benchmarks.

**Acknowledgments** C. Shen's participation was in part supported by a major grant from Zhejiang Provincial Government.

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
