# OpenReview forum: "Hierarchical  Normalization for Robust Monocular Depth Estimation"
_NeurIPS.cc/2022/Conference — NeurIPS 2022 Accept_

### Official Review · Reviewer_rDoq · 2022-06-17

**Rating:** 6
**Confidence:** 5
**Soundness:** 4 excellent
**Presentation:** 4 excellent
**Contribution:** 3 good

**Summary:**

The paper studies zero-shot monocular depth estimation. One way to improve zero-shot performance is to train on a combination of datasets. The problem with this strategy is that different datasets has different scales and shifts. One way to address this issue is to train with a scale and translation invariant loss (SSI [22]). This paper argues that the SSI loss is normalized over an entire image. A more effective way is to normalize over a small local window or a certain depth range. This paper thus propose a hierarchical depth normalization training loss, which helps their model to achieve state-of-the-art performance in zero-shot depth estimation on five major benchmarks.

**Questions:**

1. The formulation in equation (1) seems to be different from the original one in the SSI [22] paper. SSI solves for an optimal scale and shift between pred and gt depth, while equation (1) is normalizing using simple global stats such as median and mean. Why?

**Limitations:**

The authors have adequately addressed the limitations and potential negative societal impact of their work.

**Strengths And Weaknesses:**

Strength:
+ This paper proposes a simple yet very effective version of the scale-shift-invariant loss for training depth estimation network. This loss proves to help in improving zero-shot depth estimation (Table 1). I like this idea.

Weakness:
- It is not very clear if the improvement really comes from the proposed hierarchical depth loss (Table 2). The "Ours" model differs from SOTAs in the training data, the number of training samples, and the network architecture. The conclusion would be more solid if only one thing is changed at a time.  For example it is unclear if it is fair to claim that "our method outperforms previous methods by a large margin on multiple benchmarks with fewer training data than recent state-of-the-art methods, such as Leres [33] and Midas [22].", because it could be a result of using higher quality training data.

---

> ### Author Response · Authors · 2022-08-01
> **responce to questions**
>
> **Q1: It is not very clear if the improvement really comes from the proposed hierarchical depth loss**
>
> A: In Table 1 and Table A of our supplementary material, we make fair comparisons with the SOTA works, Midas[22] and Leres[33], where they correspond to the instance-level normalization methods. In the experiments, the datasets and training configurations are exactly the same, and the only difference is loss functions.
> The results show that our loss outperforms Midas by an average of 22\% on five benchmarks and outperforms Leres by an average of 20\%. Notice that we also adopt the same basic normalization operations (see Eq.1 and Eq.A) for fair comparisons, such that all performance improvement comes from hierarchical normalization.
>
>
>
> **Q2: The formulation in equation (1) seems to be different from the original one in the SSI [22] paper. SSI solves for an optimal scale and shift between pred and gt depth, while equation (1) is normalizing using simple global stats such as median and mean. Why?**
>
> Midas[22] discusses two strategies to design scale-and-shift invariant losses. One option is to align the predicted depth maps with the ground-truth depth maps based on least squares. The other strategy is to explicitly shift and scale both the predicted maps and the ground-truth maps based on the median operations. Their experiments have shown that the second option produces much better performance. Therefore, we choose the second form of SSI. Another SOTA work by Leres[33] adopts a similar strategy but replaces the median operation with mean and std for normalization. Our method can also be combined with Leres to improve the performance by up to 20\%, and the results are shown in Table A of our supplementary material.

---

### Official Review · Reviewer_ftki · 2022-07-11

**Rating:** 6
**Confidence:** 3
**Soundness:** 2 fair
**Presentation:** 3 good
**Contribution:** 3 good

**Summary:**

The authors introduce a novel strategy for monocular depth estimation that focuses on both global structure and fine-grained details. Their implementation is effective at preserving fine-grained details via the spatial and depth domains using Hierarchical Normalization. Ultimately, performing well in terms of accuracy, even against state-of Art benchmarks from before our work was published.

**Questions:**

Why the median in (1)? More elaboration on the insights could be worthwhile.

**Limitations:**

A lack of limitations in the ability to be extended to MVS; lack of demonstrating the final reconstruction results (just depth images depicted).

**Strengths And Weaknesses:**

An interesting solution to a renowned problem in monocular depth estimation.

Clearly motivated by the shortcomings of existing SOTA.

The presentation is nice, for the most part; the language is okay but should be further improved if accepted for publication.

Weaknesses:
Seems the authors ran out of space, with many means to improve figures and tables to increase the data-to-ink ratio.

Figure 3-4 : why not share x and y axes labels and make graphs easier to see in less space.

Add future work to the Limitations section.

Be consistent with style: different tables use different schemes (some use arrows, others use colors).

Avoid orphans (single words taking up entire lines, like line 205).

Lack of depth in analysis for the many ablations conducted.

---

> ### Author Response · Authors · 2022-08-02
> **responce to questions**
>
> Thanks for the constructive comments. We will carefully check and improve the organization of our paper and provide analysis in the experiments.
>
> **Q1: Lack of depth in analysis for the many ablations conducted.**
>
>
> Our supplementary material contains more experiment results and analysis to validate the generalization and effectiveness of our contributions. Specifically,
> in Table A, we combine our design with another SOTA zero-shot learning method Leres[33], and the result shows that our design can still effectively improve the performance by up to 20\%.
> In Table B, we use two more evaluation metrics to evaluate the quality of depth boundaries on iBims-1 dataset.
> We also conduct cross-domain experiments on two new synthesized datasets in Table C. We totally use 8 datasets to validate the generalization capability.
>
> Additionally,
> 1) We design a new model variant that combines our design based on the spatial domain and the depth domain, which further improves the optimal performance. (Please see our response to Q1 from Reviewer v9VC)
> 2) We also conducted experiments to validate the generalization of our design under the standard fully supervised setting. (Please see our response to Q2 from Reviewer v9VC)
> 3) We discuss a new strategy to obtain hierarchical normalization contexts, and please refer to our response to Q1 from Reviewer uzns.
>
>
>
> **Q2: Why the median in (1)? More elaboration on the insights could be worthwhile.**
>
> Previous work by Midas[22] aims to explicitly remove the shift changes between different datasets by subtracting the median. Here we adopt the same basic normalization operators for fair comparisons and ablation studies. It can also be replaced with other operations, such as mean in Leres[33]. Our design can also be combined with Leres to effectively improve performance by 20\%, as is shown in Table A of our supplementary material.
>
> **Q3: lack of demonstrating the final reconstruction results (just depth images depicted).**
>
> In Fig. A of our supplementary material, we provide the visualization of the reconstructed point clouds. We also provide demo videos that synthesize multi-view images with our depth estimation models.

---

> > ### Comment · Reviewer_ftki · 2022-08-09
> > **Rebuttal: Satisfactory**
> >
> > The authors addressed the concerns brought up in my previous review.
> >
> > Q2: It would be nice for the authors to add this snippet to the paper and elaborate more. In other words, the authors referred to prior works to justify their choice in using median, but they lack the reason why it works. Is it because of noise? Does the mean vary per change in the shift? Also, the relationship between this answer and Table A in the appendix is unclear. Please clarify in the paper; also, consider more information on this part, for it is a critical aspect of the proposed framework.
> >
> > All-in-all, excellent work!

---

> > > ### Author Response · Authors · 2022-08-09
> > > **responce to suggestions**
> > >
> > > Thanks for your suggestions. We will add this part to our manuscript and elaborate more. Yes, noise is an important reason for using median, and the mean vary per change in the shift. For example, inaccurate predictions in distant areas may constitute noise in the mean representations. When the depth values of all pixels are averaged, the relatively large depth errors in distant regions will dominate the mean representation and negatively influence the depth representations of all pixel locations. In contrast, median is less affected by these outliers.

---

### Official Review · Reviewer_uzns · 2022-07-12

**Rating:** 5
**Confidence:** 4
**Soundness:** 3 good
**Presentation:** 2 fair
**Contribution:** 2 fair

**Summary:**

This paper presents a novel normalization technique for learning monocular depth estimation networks with various sources of datasets that may have different depth scales. Unlike the previous approach that learns with the image-level normalization, which often disregards the fine-grained depth difference, this paper presents a multi-scale depth normalization that hierarchically normalizes the depth representations based on both spatial information and depth distributions. Experimental results have shown the superiority of this method even though the technique is relatively simple.

**Questions:**

- Using a hierarchical normalization has been already used in many other fields, as the authors also described, and applying this for monocular depth estimation is the first attempt in our knowledge, so it would be fine. However, due to its simplicity, I think more through experiments should be conducted. For instance, why not dividing the depth representation irregularly? or is there any way to consider additional semantic information, e.g., object boundaries, and divide the depth representation with respect to this semantic information? Without this kinds of through experiments, it would be hard to say this regular hierarchical representation is optimal.
-  Fig. 2 is hard to follow. In the left, what the authors tries to say? Maybe different colors are coming from the different spatial location? It would be better if it is clarified.
- It would be interesting if we only used a single dataset for learning monocular depth estimation networks which does not suffer from depth scale or shift problem, e.g., KITTI itself. Maybe performance is degraded with this additional local normalization layers?
- To overcome depth scale and shift problem, other loss functions may be used, for instance, AbsRel. It would be better if the other loss functions are compared.
- Computation complex may be evaluated as well.

Minor comments:
- It would be great if the authors clarify which datasets are used for each evaluation of Table 2.
- Other backbone models such as Monodepth2 can also be used? In many applications, the CNN-based models are still popularly used, so this experiment would be very interesting.

Overall I like such a simple idea, but more through evaluations would be required to argue this is optimal. I really want to see the rebuttal.

**Ethics Review Area:**

["I don’t know"]

**Limitations:**

The authors well mentioned the limitations of this paper.

**Strengths And Weaknesses:**

+ The idea of using multi-scale normalization, inspired by conventional normalization methods such as AlexNet, SIFT, and HOG, is very simple, but effective.
+ Dividing the normalization candidates as spatial and depth axis is interesting and makes sense.
+ The state-of-the-art performance is attained.

---

> ### Author Response · Authors · 2022-08-02
> **responce to questions**
>
> **Q1: ... dividing the depth representation irregularly? ...consider additional semantic information...?**
>
> As we aim to design a generic depth normalization algorithm, here we choose to define the hierarchical normalization based on regular patterns, e.g., grids, without using extra information, such that the optimization of each location is always based on multi-scale contexts.
>
> It is a good idea to incorporate additional semantic knowledge into our design. In fact, our design provides a  useful way to explicitly utilize semantic knowledge for improving predictions. For example, if we define a normalization context that covers stuff and objects based on panoptic segmentation, it emphasizes the spatial relations between different instances and stuff. On the other hand, if the normalization context is defined based on the instance masks, we can emphasize the fine-grained depth difference in object appearance, such as the structures of vehicles.
> In an application that synthesizes multi-view images of faces based on the depth prediction of portrait images, we find utilizing portrait segmentation to define the local normalization context helps the prediction of faces quantitatively and qualitatively.
>
> Here, we discuss a simple but effective variant that randomly samples pixels to define the hierarchical normalization contexts. Specifically, for HDN in the spatial domain, we randomly sample 128 regions from the image as the contexts. The lower bound of cropping size is 1/8 of the raw size, with aspect ratio restricted in [3/4, 4/3]. For HDN in the depth domain (HDN-D), we randomly define 32 depth ranges based on the ground-truth depth range, and locations being covered by a sampled range constitute a  normalization context.  The lower bound of the sampled range is set as 1/8 of the ground truth depth range in the image.
> The experiment over 5 benchmarks shows that the performance of HDNs in the depth domain and the spatial domain drops by 5% and 3%, respectively, on average, but still outperforms the baseline by 15\% and 6\%, respectively. Therefore, dividing the depth representation irregularly without extra knowledge also works, but is less effective than our initial design. Thanks for the valuable comments, and we will update these new findings and discussions in our latest manuscript.
>
> **Q2: Fig. 2 is hard to follow.**
>
> Fig.2 is used to illustrate the idea of our proposed hierarchical normalization.
> The left plot indicates a distribution of representations that may not necessarily be depth representations. (Thanks for the comments, and we should have shifted it a bit to indicate that it is not a normalized distribution.)
> In the middle, we group different representations hierarchically, as indicated by the colors, and they are normalized individually. In the depth prediction task, each dot corresponds to the depth value of a pixel location. Different colors can correspond to different normalization contexts in HDN. For example, a color means an image sub-region in HDN-Spatial. We will improve the figure illustration.
>
> **Q3: if we only used a single dataset ... which does not suffer from depth scale or shift problem**
>
> Please refer to our response to Q2 from Reviewer v9VC,  where we provide experiments to validate the effectiveness of our design under the fully supervised setting that predicts metric depth.  Our design is still very effective in this setting.
>
> **Q4:  other loss functions may be used, for instance, AbsRel. It would be better if the other loss functions are compared.**
>
> Leres[33] is the SOTA method that also designs a  scale-and-shift invariant loss.
> Section B.1 and Table A in our supplementary material show that our design can be nicely combined with it, which improves the performance by up to 20\%.
>
> Previous works[33][22] have discussed and concluded that the scale-and-shift changes between datasets can not be solved by AbsRel loss, with which the training can not converge. For example, if the gt value is small, e.g. gt=0.1, pred=1.0,  the loss will be (1-0.1)/0.1=9.0, and using AbsRel as the loss can be very unstable.
>
>
> **Q5: Computation complex**
>
> As our design is only operated on the predicted masks for computing loss, and the losses based on multi-scale contexts can be computed in parallel, the additional computation cost is negligible.  For example, the number of training iterations per second is 2.37 iter/s for the SSI baseline and 2.29 iter/s for HDN-DR, with a batch size of 16 on 4 GPUs.
>
> **Q6:... which datasets are used for each evaluation of Table 2...**
>
> Thanks for the advice. As different datasets are used in each work, we only indicate the total number of training images. We will add their detailed dataset information for clarification.
>
> **Q7:... Other backbone models**
>
> Our design is model-agnostic and can be used with any backbones. As DPT is the SOTA backbone, specifically designed for depth estimation tasks, we employ it to compare our best results with SOTAs.

---

> > ### Author Response · Authors · 2022-08-09
> > **Please let us know if you have more questions**
> >
> > thanks

---

### Official Review · Reviewer_v9VC · 2022-07-12

**Rating:** 5
**Confidence:** 4
**Soundness:** 3 good
**Presentation:** 3 good
**Contribution:** 3 good

**Summary:**

This paper proposes a normalization method for monocular-based depth estimation before computing the regression loss. It is based on scale-and-shift invariant loss. Instead of computing the median and doing the normalization globally, it split the depth map into multiple size patches and compute scale-and-shift invariant loss for each patch. They propose two strategies to generate the patches consisting of spatial domain splitting and depth domain splitting. In the zero-shot setting, the results show the model with the proposed normalization can outperform all baselines.

**Questions:**

1. Because it will compute multiscale scale-and-shift invariant loss, will the proposed loss make the training significantly slower?

**Ethics Review Area:**

["I don’t know"]

**Strengths And Weaknesses:**

1. This paper is well written. It is easy to read.
2. The proposed idea is clean and simple. We can easily apply it to other depth estimation models.
3. The improvement on the zero-shot monocular depth estimation task is significant.

Weakness:
1. Table 1 does not show the ablation when combining the different domains. It is unclear if combing all of them can get the best performance.
2. The proposed method can be applied to all monocular-based models. Have the authors tested the loss on fully supervised monocular/stereo depth models? This paper only presents the results in a zero-shot setting.
3. The proposed method is very simple, so the contribution might be limited. To improve the contribution, it is important to show the generalization of different models and datasets.

---

> ### Author Response · Authors · 2022-08-01
> **responce to questions**
>
> **Q1: Table 1 does not show the ablation when combining the different domains.**
>
> We explore three strategies to combine the design of different domains.
> 1) The first method is to directly combine the normalization contexts U (in Eq.3) generated by HDN-Depth (HDN-DR) and HDN-Spatial (HDN-S), respectively. We find that such a naive combination always produces results between the performance of two respective methods on each benchmark. The possible reason is that the hierarchical normalization contexts generated by two domains have some overlapped information, such as the global contexts. When they are averaged, these contexts are put with more weights during average,  which can not further boost the performance.
> 2) The second strategy is to take a weighted sum of the final losses (in Eq.3) generated by HDN-DR and HDN-S. By selecting multiple groups of hyper-parameters, the overall optimal result can improve HDN-DR and HDN-S across five benchmarks by an average of  0.2\% and 10.1\%, respectively.  However, we find that it does not always produce superior performance on every benchmark.
> For example, on DIODE dataset, the results of HDN-DR, HDN-S, and combination are 24.9, 36.4, and 25.3, respectively.
> This indicates that some specific benchmarks may benefit more from the hierarchical normalization in some particular domains. Therefore assigning all weights to the loss of  HDN-Depth generates the optimal performance in this benchmark.
> 3) As the hierarchical normalizations on both domains contain overlapped information, such as global contexts, the third strategy is to only select unique information for fusion. Here we simply take the local normalization contexts (the lowest two levels)  from spatial domain and fuse them with the hierarchical contexts in depth domain (HDN-DR). We find that this strategy can always produce better or comparable results across all benchmarks, which improves HDN-DR and HDN-S across five benchmarks by an average of  3.7\% and 13.3\%, respectively.
>
> We will update the detailed results into our latest manuscript. We believe there exist better strategies to combine the idea of hierarchical normalization in two domains, which will be our future works.
>
>
>
> **Q2: Have the authors tested the loss on fully supervised monocular/stereo depth models?**
>
> We conducted experiments under the standard fully supervised setting, where the goal is to predict the metric depth based on the training set of each benchmark. Specifically, we validate our design on the popular NYU V2 （*only using 795 training images in Eigen split*） dataset by adding our proposed HDN loss as an auxiliary loss to a standard L1 regression loss. The result is shown below.
> |  Loss  | NYUv2|
> |:--------:|:------------:|
> | L1  |     14.7    |
> |L1+SSI |     14.6    |
> |L1+HDN  |     **13.3**(-9.5\%)   |
>
>
> Our proposed loss can still effectively improve the performance, while the SSI baseline can hardly boost the performance, which shows the generalization capability of our methods in monocular depth estimation tasks.
> We will update a more detailed comparison and analysis in our latest manuscript.
>
>
> **Q3: it is important to show the generalization of different models and datasets.**
>
>
> Our supplementary material contains more experiment results and analysis to validate the generalization and effectiveness of our contributions. Specifically,
> in Table A, we combine our design with another SOTA zero-shot learning method Leres[33], and the result shows that our design can still effectively improve the performance, by up to 20\%.
> In Table B, we use two more evaluation metrics to evaluate the quality of depth boundaries on iBims-1 dataset.
> We also conduct cross-domain experiments on two new synthesized datasets in Table C. We totally use 8 datasets to validate the generalization capability.
>
> Additionally,
> 1) We design a new model variant that combines our design based on the spatial domain and the depth domain, which further improves the optimal performance. (see Q1).
> 2) We also conducted experiments to validate the generalization of our design under the standard fully supervised setting (see Q2).
> 3) We discuss a new strategy to obtain hierarchical normalization contexts, and please refer to our response to Q1 from Reviewer uzns.
>
>
>
>
> **Q4: will the proposed loss make the training significantly slower?**
>
> As our design is only operated on the predicted masks for computing loss, and the losses based on multi-scale contexts can be computed in parallel, the additional computation cost is negligible.  For example, the number of training iterations per second is 2.37 iter/s for the SSI baseline and 2.29 iter/s for HDN-DR, with the batch size of 16 on 4 GPUs.

---

### Meta-Review · Area_Chair_Q17U · 2022-08-22

**Recommendation:** Accept
**Confidence:** Certain

**Metareview:**

This paper addresses the problem of training a monocular depth estimation network from variable sources of data. As opposed to only using a single scaling factor as in existing work, the authors propose local schemes for normalising. While the proposed approaches are conceptually simple, they result in a non-trivial boost in performance (both qualitatively and quantitatively) and will likely be of interest in the field of monocular depth estimation.

The reviewers were broadly in support of this paper. This area-chair agress, and recommends acceptance. However, the authors are strongly encouraged to incorporate the valuable comments and suggestions from the reviewers into the revised text.

Minor comments:
* Fig 2 (a) is not clear and should be revised to make it clearer what it is trying to communicate.
* Re-title section 5.1. To “Limitations”
* Add the new results for NYUv2
* The two supplementary videos are not very informative. Should consider using different examples



**Award:**

No

---

### Decision · Program_Chairs · 2022-09-14

Accept